# Susceptibility of the Different Oxygen-Sensing Probes to Interferences in Respirometric Bacterial Assays with Complex Media

**DOI:** 10.3390/s24010267

**Published:** 2024-01-02

**Authors:** Chiara Zanetti, Liang Li, Rafael Di Lazaro Gaspar, Elisa Santovito, Sophia Elisseeva, Stuart G. Collins, Anita R. Maguire, Dmitri B. Papkovsky

**Affiliations:** 1School of Biochemistry and Cell Biology, University College Cork, Pharmacy Building, College Road, T12 K8AF Cork, Ireland; czanetti@ucc.ie (C.Z.); liang.li@ucc.ie (L.L.);; 2Cell Analysis Division, Agilent Inc., Euro House, Little Island, T45 WK12 Cork, Ireland; rafael.gaspar@agilent.com; 3National Research Council of Italy, Institute of Sciences of Food Production, Via Amendola 122/O, 70126 Bari, Italy; elisa.santovito@ispa.cnr.it; 4School of Chemistry, University College Cork, Pharmacy Building, College Road, T12 YN60 Cork, Ireland; stuart.collins@ucc.ie (S.G.C.); a.maguire@ucc.ie (A.R.M.)

**Keywords:** quenched-phosphorescence oxygen sensing, optical oxygen respirometry, bacterial cell respiration, water-soluble oxygen probes, selective media

## Abstract

Respirometric microbial assays are gaining popularity, but their uptake is limited by the availability of optimal O_2_ sensing materials and the challenge of validating assays with complex real samples. We conducted a comparative evaluation of four different O_2_-sensing probes based on Pt-porphyrin phosphors in respirometric bacterial assays performed on standard time-resolved fluorescence reader. The macromolecular MitoXpress, nanoparticle NanO2 and small molecule PtGlc_4_ and PtPEG_4_ probes were assessed with *E. coli* cells in five growth media: nutrient broth (NB), McConkey (MC), Rapid Coliform ChromoSelect (RCC), M-Lauryl lauryl sulfate (MLS), and Minerals-Modified Glutamate (MMG) media. Respiration profiles of the cells were recorded and analyzed, along with densitometry profiles and quenching studies of individual media components. This revealed several limiting factors and interferences impacting assay performance, which include probe quenched lifetime, instrument temporal resolution, inner filter effects (mainly by indicator dyes), probe binding to lipophilic components, and dynamic and static quenching by media components. The study allowed for the ranking of the probes based on their ruggedness, resilience to interferences and overall performance in respirometric bacterial assays. The ‘shielded’ probe NanO2 outperformed the established MitoXpress probe and the small molecule probes PtGlc_4_ and PtPEG_4_.

## 1. Introduction

Measurement of bacterial cell respiration allows for the monitoring of microbial growth and metabolism, of the effects of drugs and antimicrobial treatments, of microbial contamination and spoilage of food and environmental samples and of sterility, toxicity, etc. [1,2,3,4]. Optical-oxygen-sensor-based respirometry is well suited for this, and has been used in food and pharmaceutical industries, biomedical research, and environmental science, particularly for the high throughput analysis of total aerobic viable counts (TVCs) and determination of EC50 values [2,3,4]. This technique provides a more rapid, simple, and convenient alternative to conventional agar plating [2,4] and other methods that are more laborious, time-consuming, and poorly automated [5,6].

Quenched-phosphorescence sensing systems allow facile quantification of dissolved O_2_ concentration in biological samples and the measurement of the respiration activity of both mammalian [7] and bacterial [1,2] cells on widely available spectroscopic equipment, such as multi-label plate readers [1,2,4]. Depending on the type of oxygen-sensitive material, assay vessel, detector and samples used, respirometric assays form several categories. One common platform, here called macro-respirometry, employs solid-state sensor coatings based on near-infrared Pt-benzoporphyrin dye [8,9], deposited inside disposable plastic tubes (2 mL, 15 mL, or 50 mL volume) and read with a handheld sensor reader such as Optech (Mocon/Ametek) or Firesting-GO2 (Pyroscience) [8,9]. Another platform, called micro-respirometry, uses a red-emitting O_2_ sensing probe based on Pt-coproporphyrin dye (PtCP), which is added directly to liquid samples and measured in standard 96/384-well plates on a benchtop time-resolved fluorescence (TR-F) reader [2,3,7]. The MitoXpress-Xtra (MitoXpress) probe used in this platform has convenient spectral and lifetime characteristics and compatibility with various TR-F readers. This probe provides good sensitivity and signal-to-noise ratio, stable and calibration-free phosphorescence lifetime-based sensing of O_2_. MitoXpress has been validated in many bioanalytical applications and is currently regarded as the gold standard in micro-respirometry [3].

O_2_ sensor-based respirometric assays have been successfully used with different bacteria (*E. coli* and others) in simple-growth media, such as nutrient broth (NB) [3]. However, in more complex media that are used for the selective determination of bacteria, micro-respirometry assays with a MitoXpress probe did not perform well and were superseded by macro-respirometry assays with solid-state sensors [9]. Thus, in some selective media that contain indicator dyes, chromogenic substrates, surfactants, or other additives for the enrichment of certain bacteria [10], MitoXpress probes are susceptible to strong interferences.

On the other hand, new O_2_ probes, structurally different but spectrally similar to MitoXpress and which can potentially improve performance and ruggedness of micro-respirometry assays, have been described in recent years [11,12,13]. In this study, we investigated four different soluble O_2_ probes based on Pt-porphyrin phosphors, including the benchmark MitoXpress probe, in model micro-respirometry assays with *E. coli* cells and five growth media with complex compositions and optical properties [9]. We aimed to identify the best performing probe for such assays and elaborate on the causes of interference on probe signals and respiration profiles.

## 2. Materials and Methods

### 2.1. Materials

Thiamine, pantothenic acid, nicotinic acid, bile salts, ferric ammonium citrate, “Lab-Lemco”, peptone, yeast extract, sodium lauryl sulfate, (5-bromo-4-chloro-3-indolyl-beta-D-galacto-pyranoside (X-gal), 4-methylumbelliferyl-β-D-glucuronide (MUG), isopropyl ß-D-1-thiogalactopyranoside (IPTG), neutral red, phenol red, bromocresol purple, lactose, and sorbitol were acquired from Sigma-Aldrich (Burlington, MA, USA). Nutrient broth (NB) was acquired from Fisher Scientific Oxoid (Dublin, Ireland), selective and differentiating media specific for *E. coli* including the MacConkey (MC), Rapid Coliform ChromoSelect (RCC), M-lauryl sulfate (MLS), and Minerals-Modified Glutamate (MMG) broths, were prepared as described in [9]. Heavy mineral oil was acquired from Cargille Laboratories. All other chemicals and solvents were of analytical grade and solutions were prepared from Milli-Q water (Millipore, Carrigtwohill, Ireland) or sterile DMSO (Sigma-Aldrich). MitoXpress-Xtra (MitoXpress) probe was acquired from Agilent. NanO2 probe was synthesized in our lab as described in [12], and PtGlc_4_ and PtPEG_4_ probes as described in [11,14].

### 2.2. Respirometric and Turbidimetric Assay Procedures

*Escherichia coli* (*E. coli*) cells, strain NCIMB 11943, were obtained from the School of Microbiology, University College Cork and stored in LB broth containing 80% glycerol at −80 °C. Fresh working stock of *E. coli* was prepared by overnight incubation in NB as described in [4,9], from which 1:10 serial dilutions in each medium were made. Then, O_2_ probes were added to produce final *E. coli* concentrations of 10^4^ and 10^6^ CFU/mL and MitoXpress (0.3 µM), NanO2 (0.3 µM), PtGlc_4_ (3 μM) or PtPEG_4_ (3 μM). Then, 200 µL aliquots of each solution were dispensed in triplicates into wells of a 96-well plate (Sarstedt). Solutions of the probes in sterile media without bacteria (negative controls) and *E. coli* dilutions in media without O_2_ probe were also prepared, dispensed on the plate, and measured as blank signals. Sample wells were then sealed with 70 μL of mineral oil and the plate was placed in a Victor2 (PerkinElmer) reader equilibrated at 37 °C and monitored in time-resolved fluorescence (TR-F) mode for 10 h, measuring signals in each well every 5 min, under the following settings: excitation filter—340 nm, emission filter—642 nm, two delay times—30 μs and 70 μs, and gate time—100 μs (for each delay). Phosphorescence lifetime values (LT) were calculated using the formula for rapid lifetime determination (RLD): LT = (t_1_ − t_2_)/ln(F_1_/F_2_), where t_1_ and t_2_ are the delay times, and F_1_ and F_2_ are the corresponding intensity signals [2,4,7]. The resulting LT values were plotted over time to produce respiration profiles for each well, media and probe controls. For PtGlc_4_ and PtPEG_4_ probes, t_1_ and t_2_ were set at 25 μs and 50 μs, respectively [11,14].

Profiles of absorbance/light scattering were also measured on a Victor2 reader in a plate with samples of *E. coli* in the different media (no probe) and negative controls (media, no cells), using 600 nm and 405 nm filters and the same temperature and kinetic settings as for TR-F measurements.

### 2.3. Phosphorescence Quenching Experiments

Phosphorescence quenching experiments with individual media components were carried out on a Cary Eclipse fluorescence spectrometer as follows. Compounds were prepared as concentrated 20× stock solutions in water or DMSO. They were added stepwise to a standard 10 mm quartz cuvette containing 2 mL of working dilution of the probe in NaCl (5.0 g/L), to produce 0.5×, 1× and 2× concentrations (where 1× is the compound concentration in the media). Before and after each addition, phosphorescence spectra were recorded (excitation 390 nm, emission 600–700 nm), and after the last compound addition phosphorescence decay was also measured (Exc390 nm/Em650 nm). The resulting Int and LT data were analyzed to determine the Stern–Volmer quenching constant, K_SV_ [15], and the % of quenching at 1× concentration. Sample absorbance spectra (300–600 nm) were also recorded and analyzed for possible interferences on the TR-F and lifetime signals.

### 2.4. Imaging of Phosphorescently Stained E. coli Cells

*E. coli* cells, 10^6^ per mL, were incubated with the probes at standard working dilutions in NB for 3 h at 37 °C. Cells were pelleted at 3000 rpm for 10 min, the supernatant was removed, and pellets were resuspended in 100 μL of NB containing the probe. Small aliquots were dispensed on a MilliCell Disposable Haemocytometer and imaged in phosphorescence intensity and PLIM mode on the confocal microscope at room temperature under 405 nm laser excitation and 40× magnification lens.

## 3. Results and Discussion

### 3.1. Selection of the Cell Model and Media

*E. coli* is a classical bacterial cell model, frequently used in non-selective assays such as the total aerobic viable counts (TVCs) performed by plating on solid media (ISO 4833-1:2013 agar plating method [16]). It is also used in conventional assays with selective media with enrichment and/or predictive identification of bacterial groups or families such as coliforms, *Enterobacteriaceae* [17,18], or pathogenic species [19,20]. *E. coli* cells have been extensively validated in O_2_ sensor-based respirometric assays, e.g., in rapid TVC testing of crude food samples in liquid media [4,8,21]. Various non-selective and selective media are used with *E. coli* cells in different applications [9]. We have chosen the five media used in the previous study with the solid-state O_2_ sensor coatings and MitoXpress probe [9]: nutrient broth (NB), McConkey (MC), RCC, MMG and MLS, which each have different compositions (Appendix A, ESI) and contain basic nutrients and special additives, such as indicator dyes, metabolites, surfactants, vitamins, substrates. These components promote the growth of target microorganisms while suppressing unwanted species, or report on the presence of target microorganisms by color changes or other means. Due to the different composition and optical properties (Figure 1), these media are expected to act differently on the O_2_ probes and corresponding respirometric assays.

### 3.2. Selection and General Comparison of the O_2_ Probes

MitoXpress probes are well established and routinely used in micro-respirometry with mammalian and bacterial cells [2,3,7,9], normally at a working concentration of 0.3 µM (with respect to the dye, Table 1). They comprise a macromolecular structure produced by covalent linkage of an amino-reactive phosphor, PtCP-NCS, with a water-soluble polypeptide carrier, BSA (Figure 2) [2,3,7]. PtCP phosphor has high emission yield (Φ = ~0.4, unquenched) and specific brightness (ε*Φ = 34,000 M^−1^cm^−1^) [22]. Upon conjugation to BSA its brightness in aqueous buffers becomes reduced several fold due to static quenching [7], but MitoXpress still produces high Int signals and stable LT values on different instruments and produces smooth and meaningful respiration profiles in various cell-based assays, with modest consumption of the reagent [3,7].

The other three probes comprise the PtPFPP phosphor placed in different micro-environments: an amphiphilic polymer for the NanO2 or aqueous medium for the small molecule probes (Figure 2). While PtPFPP is hydrophobic and ~5 times less bright than PtCP [22], it is readily available, inexpensive, and amenable to simple click-modifications with thiols [23,24,25]. Thus far, these probes have not been assessed in micro-respirometry assays.

NanO2 was developed for sensing intracellular O_2_ and phosphorescence lifetime imaging microscopy (PLIM) [12]. It comprises an aqueous dispersion of cationic nanoparticles of an amphiphilic RL-100 hydrogel, with PtPFPP molecules physically trapped in their hydrophobic core (Figure 2B, Table 1). The polymer ‘shields’ the phosphor from external environment and interferences by sample components, and provides optimal quenching and sensitivity to O_2_. The nanoparticles have an average size of 35–40 nm and positive surface charge. They can penetrate and passively stain various mammalian cells and tissue samples by simple addition to the medium and incubation [18]. NanO2 can be stored for many years (as concentrated stock in water, at 4 °C); however, its colloidal and structural stability in different environments has not been studied in detail.

PtGlc_4_ is a hydrophilic small-molecule probe with cell-penetrating ability, designed for intracellular use, deep tissue staining and the mapping of O_2_ concentration by PLIM. It is produced by click-modification of PtPFPP with four β-D-thio-glucose moieties (Figure 2C) [11]. Compared with MitoXpress and NanO2, PtGlc_4_ shows a high degree of internal quenching of the PtPFPP phosphor [11]. PtPEG_4_ is another derivative of PtPFPP, with four thiolated hexa-PEG peripheral chains (Figure 2C). PtPEG_4_ is less hydrophilic than PtGlc_4_ and is cell-impermeable [14].

### 3.3. Basal and Maximal Phosphorescent Signals of the Probes in the E. coli Assays

In standard non-selective NB medium, a MitoXpress probe produces stable baseline with mean basal LT of ~23 µs, which increases to ~75 µs upon sample deoxygenation caused by bacterial growth (Table 2). Such LT and spectral characteristics of MitoXpress (excitation at 360–400 nm, emission at 640–670 nm, respectively) are suitable for its detection on standard TR-F readers. The latter usually employ Xe-flash lamp excitation (pulse duration ~20 μs) and time-gated detection of emission with a photon counting PMT, which provide high intensity signals and signal-to-background (S:B) ratio, with fast and accurate determination of the probe LT by the rapid lifetime determination (RLD) method [26].

The NanO2 probe, when used at the same working concentration as MitoXpress (0.3 μM) in NB, produces ~6-times higher Int signals (1.75M cps vs. 240 k cps) and similar LT values: 26.9 µs and 70.8 µs in air-saturated and deoxygenated conditions, respectively (Table 2). Its long LT is attributed to the rigid micro-environment of PtTFPP molecules inside the nanoparticles, which decreases non-radiative quenching from vibronic coupling [15].

PtGlc_4_ showed a high degree of internal quenching, which reduces its Int and LT signals [11]: 160 k cps and 10.3 µs in air-saturated and 7.50 k cps and 17.1 µs in deoxygenated conditions, respectively (Table 2). These signals correspond to 3 μM concentration, i.e., 10-times higher than for MitoXpress and NanO2 (Table 1).

PtPEG_4_ also showed significant internal quenching. At 3 μM concentration, its Int and LT signals in NB were 26 k cps and 11.3 µs in air-saturated and 180 k cps and 29.1 µs in deoxygenated conditions, respectively (Table 2).

From the data in Table 2, the following conclusions can be drawn:For the MitoXpress probe, basal intensity signals in the RCC and MLS media (highlighted in red) were strongly quenched, dropping below the acceptable levels (>30 k cps) for this type of instrument and assay [27]. On the other hand, its maximal Int signals remained high and easy to measure in all of the media. Because LT in RCC and MLS reduced considerably, this required customized LT thresholds to be used for the determination of TT values. Overall, the usability of the MitoXpress probe in RCC and MLS media was somehow problematic.The NanO2 probe retained high basal and very high maximal Int signals in all of the media, despite their 3–6-fold quenching in MMG and MLS media. LT signals also remained stable but moderately quenched in MMG—the only problematic media for NanO2.The PtGlc_4_ and PtPEG_4_ probes produced low basal Int signals in NB and MLS, and their basal LT signals were <13 µs in all of the media. Such short LT of PtGlc_4_ and PtPEG_4_ resulted in large losses of Int signals measured at standard delay times of 30 µs and 70 µs. When t_1_ and t_2_ settings were changed to 25 µs and 50 µs, blank signals largely increased and S:B ratio decreased for these probes. Lowest basal Int signals for these probes were seen in NB and MLS, and highest in RCC. Despite their similar structure, PtGlc_4_ produced much higher maximal Int signals than PtPEG_4_, and different pattern of signals in the media. Thus, under standard respirometric assay settings, both PtGlc_4_ and PtPEG_4_ probes are barely usable in all of the media.

### 3.4. Full Respiration Profiles of E. coli Produced with Different Media and Probes

Phosphorescent signals of an O_2_ sensor have a reciprocal relationship with sample O_2_ concentration, described by the Stern–Volmer equation [3,7,15]. As a result, in respirometric bacterial assays, O_2_ probes produce characteristic sigmoidal profiles of the intensity (Int, cps) and lifetime (LT, µs) signals. Initially, the signal produces a flat baseline, which corresponds to air-saturated conditions in the sample with maximal quenching of the probe by O_2_. Then, at a certain time, the signal undergoes steep transition from its low baseline to high plateau levels. This reflects the change from oxygenated to deoxygenated conditions in the sample, due to rapid exponential growth and increase in cell density. The time of this transition is governed by the initial concentration of bacteria in the sample, the type of growth media and assay conditions [2,3]. Bacterial cell respirometry usually monitors primary sensor signals (Int or LT). The LT-based mode provides reliable determination of TT values (h) for respiring samples. It allows simple enumeration of bacteria in unknown samples: adding the sample to growth medium, monitoring its respiration profile, and determining the time to reach signal threshold (TT, hours) which is set above the baseline. Then, measured TT values are converted into cell concentrations (CFU/mL or CFU/g) by applying a known calibration equation [2,8,9]. However, this basic theory does not account for possible optical effects and interferences associated with the medium, cells or probe used. O_2_ concentration readout is more prone to measurement errors and artefacts than an LT-based readout [3]. Especially in the challenging conditions used here.

Figure 3A shows full respiration profiles of *E. coli* for the different probes in simple non-selective NB media. NB contains only NaCl, peptone (source of organic nitrogen, amino acids, and long chain fatty acids), yeast extract (source of carbohydrates, vitamins, other organic nitrogen compounds and salts) and no additives [15]. It appears that classical respiration profiles are recognizable in NB only for the MitoXpress (Int and LT) and NanO2 (LT) probes. The Int signal of NanO2 showed a downward drift in the plateau region and a small upward drift of the baseline, which can be attributed to probe binding to surfaces or media components. Nevertheless, NanO2 provides the reliable and accurate determination of TT values, so it remains usable. Conversely, PtGlc4 and PtPEG4 probes produced unusual bell-shaped Int profiles and non-classical LT profiles, with marked upward drifts of the Int baseline and unstable LT baseline (Figure 3A). We attribute these effects to the low LT values and S:B ratio, which make these assays susceptible to interferences and changes in sample optical properties (absorption, scattering) during the assay (see below).

MC medium resembles NB, with yeast extract replaced by bile salts and lactose, and with neutral red indicator (pKa = 6.9), which changes color from orange to magenta (Figure 1). Int signals for all of the probes in MC were 1.5–2-times lower than in NB. Again, PtPEG_4_ and PtGlc4 showed unacceptably low baseline Int and LT (<15 µs) signals (Table 2) and skewed respiration profiles (Figure 3B).

RCC medium contains peptone, sorbitol and salts, sodium lauryl sulphate (anionic surfactant, 0.1 g/L) and UV chromogenic substrates X-Gal and MUG. In colorless RCC, basal and maximal LT signals of NanO2 were close to normal, but some odd spikes occurred at the start of the plateau region. The MitoXpress probe showed very low basal intensity signals, reduced basal and maximal LT signals. We attribute these effects to the denaturing action of SLS on the BSA carrier. PtGlc_4_ showed bell-shaped profiles and low LT signals (Table 2, Figure 3C). Response to sample deoxygenation and TT values were detectable, though with difficulties for some of the probes.

Figure 3D shows Int and LT profiles of E. coli cells in MMG medium, which contains a Fe^2+^ salt, essential vitamins (pantothenic acid), aromatic compounds (thiamine, nicotinic acid), and bromocresol purple pH indicator which, in the presence of E. coli, turns media color from pale blue to light yellow [9] (Figure 1). In MMG, baseline and maximal Int signals of all of the probes are reduced 3–6 fold. For PtPEG_4_, baseline Int signals dropped below the acceptable threshold for respirometry (>30 k cps) [26,27]. Only the NanO2 probe showed close-to-normal sigmoidal profiles of Int and LT signals, but its LT baseline reduced to 15 µs and maximal LT down to 30 µs (Table 2). LT profiles of MitoXpress were close to sigmoidal, but Int profiles were bell-shaped. PtPEG_4_ also showed bell-shaped Int profiles, complex LT profiles, and lowered and drifting up Int and LT baselines. Nevertheless, response to sample deoxygenation was visible for all of the probes, except PtGlc_4_. In addition, signal onset times (or TT values) in MMG medium were 0.2–0.4 h longer than in NB (Table 2), reflecting a shift between 10^4^ and 10^6^
*E. coli* CFU/mL. These results suggest that bromocresol purple indicator, or other components of MGG, quench the probes’ signals and thus alter the shape of their respiration profiles.

MLS medium contains 1 g/L of SLS surfactant, i.e., 10 times more than RCC, and phenol red indicator (pKa = 8.0) which changes color from red to orange. MLS is likely to denature the MitoXpress probe, thus reducing its basal Int signal and making it difficult to use in respirometric assays. PtGlc_4_ and PtPEG_4_ are also strongly quenched and unusable in MLS. Only NanO2 retained sufficient brightness and appropriate span of LT changes, which are shifted down. From its lifetime profile it was possible to calculate TT value, though upper Int and LT signals were not so stable and flat (Table 2, Figure 3E).

### 3.5. Analysis of Blank Signals and Spectroscopic Effects of the Media

The observed effects on the probe signals and respirometric profiles of *E. coli* cells in the different media can be due to the inner filter effects (IFE), or to static and dynamic quenching by media components [15]. However, the bumps on signal profiles, drift of the baseline, and bell-shaped and other unusual signal profiles are more difficult to explain. These effects can have instrumental (light scattering, high blanks, low S/N, inaccurate LT calculation, etc.), biological (metabolic switching, changes of cell density, growth rate), or chemical origin.

To explain these effects, we first measured blank Int signals for the samples containing different media, with or without cells, and with no probe, under the settings used in the respirometry. Such blanks, measured at a delay time of 30 μs, appeared to be high: ~3 k cps for the MMG, ~7 k for the MLS and ~15 k cps for the MC (Figure 4A,B). When delay time was reduced to 25 μs, blank signals increased more than 3 fold (Figure 4C,D). Thus, such optimization of the measurement conditions for the small molecule probes is not efficient. On the other hand, flat profiles of the blank samples suggest that probe signals are not influenced by changing cell density (largely increases over 10 h), and sample deoxygenation which occurred at ~2.5 h for Figure 4A,C (refer to Figure 3A–E). Moreover, introducing blank correction in the RLD method of LT calculation did not improve the resulting respiration profiles, which kept the same shape for NanO2 and MitoXpress (not shown) and strange shape for PtGlc_4_ and PtPEG_4_ (see exemplary profiles in Appendix A, ESI). Thus, high blanks only partly explain the unusual effects on the respiration profiles of tested probes and media.

Next, we carried out densitometry with the same samples as in O_2_ respirometry but measuring absorbance profiles at 405 nm and 600 nm (Figure 5). The first wavelength is close to the excitation band (300–380 nm) of the probes, while the second is close to their emission. Such measurements were expected to reveal the effects of changing cell density and light scattering during the assay, as well as color changes in the media induced by metabolizing cells. Notably, A_600_ densitometry is also used to enumerate bacteria [28,29].

The A_405_ profiles of *E. coli* in the different media (Figure 5A) reveal the following:Baseline A_405_ values and changes were rather low (0.12–0.3 AU) for the NB, MMG and RCC; moderate for MC (0.5–0.9 AU); and very high and practically unusable for MLS (A_405_ > 3 AU). This explains the large attenuation of probe Int signals in MLS and moderate attenuation in MC, due to the prominent inner filter effects for these media.A_405_ profiles in MMG and RCC had a sigmoidal shape, while a bi-phasic shape with a linear terminal part was observed in NB and MC.Compared with the corresponding respiration profiles, signal onset (TT) occurred later in the densitometry, and subsequent signal changes lasted longer (Figure 5, Table 3).

The A_600_ profiles in NB, MC and RCC (Figure 5B) were similar in shape to A_405_, while MMG had an inverted shape with no changes occurring in the first 5 h. Again, A_600_ profiles reflected mostly changes in media color, rather than in the biomass and light scattering. Notably, A_600_ profiles in MLS were measurable, had a sigmoidal shape and long TT values (>4 h). The observed differences with respirometry data are understandable. While both assays report on cell growth and the number of bacteria, their measured parameters—dissolved O_2_ for respirometry and light scattering and absorption for densitometry—have a different relationship with CFU numbers [2].

The main contributing instrumental factors identified so far have been high blank signals and low S:N ratio, due to light scattering and short-lived autofluorescence and to Xe-flash lamp afterglow that limits the temporal resolution of the TRF/RLD to >15 μs [30,31]. The optical effects of selective media are mainly due to the broad absorption bands of the different indicator dyes and spectral changes taking place during the assay [9]. The indicators produce strong inner filter effects (IFE), attenuating probe excitation at around 390 nm and emission at 650 nm. These effects are more specific to the media; however, probes producing low S:B ratio could be more affected than those with high S:B. At the same time, certain media components can also have interfering effects on the probes and assays.

### 3.6. Analysis of Individual Media Components for Interferences

The complex composition of the selective media (Appendix A) points to possible interfering action by some components on probe signals. To identify the main interfering components and their mode of action, we carried out fluorometric titrations of MitoXpress, NanO2 and PtGlc_4_ probes at working concentrations in 5.0 g/L NaCl with individual components of the media (Appendix A). Absorption spectra and changes in probe Int and LT signals upon addition of the compound were measured in a cuvette, and % of signal change at the concentrations used in the media were calculated (Table 4).

As expected, simple media components, such as salts (NaCl, phosphates) and small-molecule metabolites (saccharides, vitamins, etc.) did not show any significant interference on the probes. On the other hand, some more complex ingredients (peptone, yeast extract), colored or chromogenic components (pH indicators, enzyme substrates) and special additives (surfactants, metal ions) did have an effect. Thus, in addition to its strong Int quenching, bromocresol purple also reduced LT of the NanO2 (12%), MitoXpress (32%) and PtGlc_4_ (12%), thus pointing to dynamic quenching or FRET [32].

Peptone and yeast extract showed significant absorbance and IFE on the probes, while Lab-Lemco, peptone and yeast extract enhanced (by up to 436%) the Int signals of PtGlc_4_ and MitoXpress probes, without affecting their LT values (Table 4). Peptone also showed strong fluorescence with maximum at 560 nm (Appendix A), which resembles that of riboflavin [33].

Because the ‘shielded’ NanO2 probe was least affected, we attribute these effects to binding interactions between the probe phosphor and lipophilic components present in peptone and yeast extract, which can also prevent stacking interactions and/or phosphor aggregation in aqueous solution. The latter is supported by the effects of bile salts (lipophilic derivatives of cholesterol), which largely enhanced the Int signals of PtGlc_4_ (826%), MitoXpress (62%) and NanO2 (31%) probes, with little effect on the LT. Bile salts are known to form micelles and solubilize hydrophobic molecules [34].

Sodium lauryl sulfate, an anionic surfactant, largely enhanced the Int signal of PtGlc_4_ (268%), moderately quenched MitoXpress (21%) and had a marginal effect on NanO2. Because quenching of the LT was significant (50–16%—see Table 4), we attribute these effects to phosphor solubilization by the SLS and denaturation of BSA carrier, which in turn, changed the phosphor’s micro-environment and K_SV_.

Ferric ammonium citrate (used for the identification of *Enterobacteriaceae*) slightly quenched the Int and LT signals of PtGlc_4_ (8.5/7%) and MitoXpress (4/12%), while NanO2 was unaffected. The quenching could be due to the heavy atom and paramagnetic effects of the Fe^3+^ ions [35] and the tendency of phenyl groups to coordinate Fe^3+^ [36].

MUG (4-methylumbelliferyl-β-D-glucuronide), a fluorogenic substrate of β-galactosidase, absorbs below 400 nm and, when hydrolyzed, produces blue fluorescence under 365 nm excitation. X-Gal (5-bromo-4-chloro-3-indolyl-beta-D-galacto-pyranoside), a cell-permeable chromogenic substrate of β-galactosidase, when hydrolyzed, produces blue precipitate (A_max_ at 615 nm) and fluorescence at 600–700 nm. X-Gal and MUG quenched the Int of all the probes by 16–17%, with a minor quenching of the LT by MUG (3.7–6.7%), but not X-Gal.

All other compounds, including thiamine (vitamin b1, co-factor for essential enzymes), pantothenic acid (vitamin B5, required for fatty acid biosynthesis) and nicotinic acid (growth factor) were seen to alter probe signals only marginally. Of these, only thiamine was reported as a quencher for o-phenanthroline derivatives [37].

### 3.7. Cell Penetration and Staining Ability of the Probes

Cell-penetrating ability of different compounds, including O_2_ probes, is determined by many factors, which relate to the cell, medium and compound in question. Such factors have been investigated in detail, but mainly with mammalian cells. Thus, positive charges and saccharide substituents (e.g., glucose, galactose) on the compound/probe tend to promote compound internalization [11,12,38], whereas PEGylation prevents it [14]. However, these rules are not universal and for each system theoretical predictions usually require tedious experimental studies to optimize probe structure and elucidate the mechanisms of cellular uptake. Moreover, the cell-penetrating behavior of an O_2_ probe determined with mammalian cells cannot be simply extrapolated on bacterial cells, as they have very different membrane compositions, transport mechanisms, growth rates, etc.

Of the four probes, NanO2 and PtGlc_4_ are known as cell-permeable for mammalian cells, allowing them to passively stain, sense and image intracellular O_2_ concentrations [11,12]. In contrast, MitoXpress and PtPEG_4_ probes are regarded as cell impermeable, as they poorly stain mammalian cells [7,14]. However, the cell penetrating ability and phosphorescent staining efficiency of all these probes with respect to bacterial cells have not been studied so far.

Using confocal fluorescence microscopy [11,12], we performed phosphorescent staining and imaging experiments with *E. coli* cells and determined cell-staining behavior of the probes in conditions of respirometric assays. Phosphorescence intensity images in Figure 6 reveal that, after 3 h incubation with *E. coli* (10^6^ CFU/mL) in NB at 37 °C, only the NanO2 probe provided efficient phosphorescent staining and visualization of the cells. The other probes showed no cell staining (MitoXpress) or very few stained cells in the field of view (more cells for PtPEG_4_ than for PtGlc_4_).

Thus, unlike with mammalian cells [11], PtGlc_4_ did not show good staining of *E. coli*, and behaved as a cell-impermeable probe. MitoXpress and PtPEG_4_ behaved as extracellular, cell-impermeable probes, i.e., as with mammalian cells (see Table 1, [12,13]).

### 3.8. Overall Comparison and Ranking of the Probes

The above results reveal that a number of components present in the selected media can influence probe signals and respiration profiles. The observed effects range from 86% quenching to 826% enhancement of basal Int signal and up to 55% quenching of the LT. IFE can also occur (Table 4). Note that, if a media contains several such components, the overall effect will be additive. While such interferences can be tolerated by MitoXpress and NanO2, PtPEG_4_ and PtGlc_4_ probes with low S:N ratio and low ‘spare’ sensitivity cannot operate reliably. These produce unusual respiration profiles, which are difficult to interpret and use for the enumeration of bacteria via TT determination [9].

Such quenching interferences and attenuation of probe Int signals can be counter-acted by: (i) using higher probe concentrations; (ii) choosing brighter probes, such as NanO2; (iii) excluding the main interfering components, provided they are not critical for the intended action of the medium; and (iv) choosing alternative or new media with different composition but similar biological end result and lower interferences. However, these strategies have their own limitations (e.g., probe/media availability, assay costs).

The four Pt-porphyrin-based probes of different type (small molecules, macromolecular conjugate and core–shell nanoparticle structures, Figure 1) produced different interference patterns in the assays with the chosen media (Table 4). The data collected allows for the ranking of the probes based on their behavior and analytical performances.

Comparative analysis of the respiration profiles in the different media (Figure 2, Figure 3 and Appendix A), blank samples and absorption profiles (Figure 5 and Figure 6), and Table 4 data, puts NanO2 probe on top of the list. NanO2 shows consistently lower interferences than the other probes (Table 4), it operated reliably in four out of the five media, including the most challenging RCC and MLS, but had some issues with RCC (Table 5). The superiority of NanO2 is due to its ‘shielded’ structure, such that phosphor molecules in the nanoparticle core are not accessible for sample/medium components. The cell-penetrating ability of NanO2 with bacteria is also beneficial, as the cells further shield the probe from interfering compounds in the media. Therefore, NanO2 replaces MitoXpress probe, which is currently used in bacterial assays and regarded as the ‘gold standard’.

The benchmark probe, MitoXpress, although it worked reliably in the NB and MC, failed in RCC and MLS, and to some degree in MMG. Thus, it was ranked second. Additionally, the PtGlc_4_ and PtPEG_4_ small-molecule probes both performed poorly, even when used at 10-times higher concentrations. PtGlc_4_ still worked, though with problems, in the NB, MC and RCC, but failed in MMG and MLS, so was ranked third and the outsider PtPEG_4_ worked only in NB and RCC. The main limiting factors for PtPEG_4_ and PtGlc_4_ were their short lifetimes, below-instrument temporal resolution, low brightness, and hydrophilicity. While other O_2_ probes are available (e.g., fluorescent Ru(II) or Ir(III) complexes [39,40,41]), their LT and/or spectral characteristics are poorly suited for respirometric assays on standard TRF readers.

## 4. Conclusions

General usability and performance of the four different types of soluble O_2_ probes in respirometric bacterial assays were assessed using *E. coli* cells and five complex media designed for selective bacterial assays. The detailed spectroscopic analysis of this model has revealed multiple interfering and limiting factors, which include the instrumentation used, probe structure and photophysical characteristics, and action of particular media components on the probes.

Regarding instrumental factors and probe characteristics, short LTs of the probe (<15 μs) fall below the instrument temporal resolution, limited by the Xe-flash lamp pulse duration [9]. This leads to large measurement errors and inaccurate LT determination by the RLD method [30]. Similarly, high blank signals producing low S:B ratio for the ‘dim’ PtGlc_4_ and PtPEG_4_ probes, lead to unstable measurements and LT-based O_2_ sensing, with respiration profiles of unusual shape. The asynchronous changes in sample absorbance, light scattering and probe Int signal, especially when combined with low S:B ratio, may also result in weird bumps and respiration profiles for some probes and media.

A number of individual media components were seen to interfere with the probes, and such interferences were probe-specific. They can be categorized based on their mode(s) of action as follows:Colored substances acting via inner filter effect or FRET, such as pH indicators, chromogenic substrates (MUG, X-gal), and other UV absorbers;Surfactants and lipophilic molecules (e.g., lauryl sulphate, bile acids) which can denature protein-based probes, solubilize the phosphor in small molecules or disturb the nanoparticle structures, thus changing phosphor’s micro-environment and O_2_ sensitivity, seen as quenching or enhancement of probe signals;Complex ingredients of biological origin (peptone, yeast extract, Lab-Lemco), which have similar effects as the previous group, mediated by probe binding;Compounds containing heavy atoms (X-Gal, Ferric ammonium citrate) that may cause dynamic or static quenching; and reducing or oxidizing agents that can modify phosphor’s structure and environment.

Comparative analysis of respiration profiles produced by the different probes and media allows for the ranking of the probes, based on their ruggedness, resilience to interferences and operational performance. This puts the ‘shielded’ NanO2 probe on top of the list, as it showed consistently lower interferences and operated reliably in four out of the five complex media, including the most challenging RCC and MLS media, with some problems seen in RCC. The ‘gold standard’ MitoXpress probe was ranked second, as it was usable only in two or three out of the five media. The small molecule probes PtGlc_4_ and PtPEG_4_ were the least usable, being hampered by their low brightness, low basal LT signals, and stronger interactions with media components.

Thus, the order of probe performance is NanO2 > MitoXpress > PtGlc_4_ > PtPEG_4_. Similarly, the five media can also be ranked based on their ‘harshness’ with respect to the O_2_ sensing probes, as follows: MLS > MMG > RCC > MC > NB. Consideration of these factors and effects of the different sample/media components aids the selection of appropriate probe and media for new applications, and the development of improved probes for O_2_ respirometry.

## Figures and Tables

**Figure 1 sensors-24-00267-f001:**
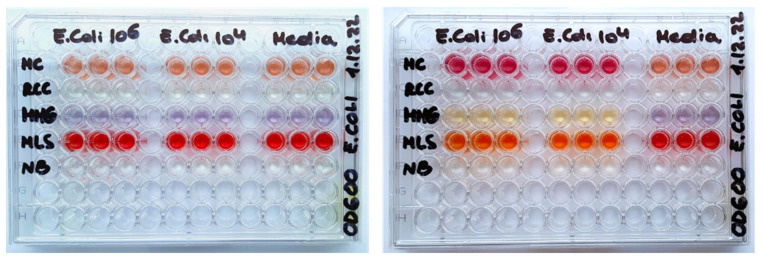
Photographs of 96-well plates with the different media and *E. coli* loads (zero, 10^4^ or 10^6^ CFU/mL), before (**left**) and after (**right**) an overnight incubation at 30 °C.

**Figure 2 sensors-24-00267-f002:**
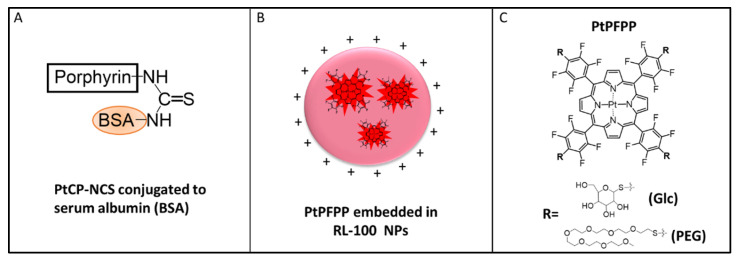
Schematic representation of the O_2_ probes used: (**A**) MitoXpress, comprising a macromolecular conjugate of bovine serum albumin (BSA) and PtCP-NCS; (**B**) NanO2, comprising core–shell nanoparticles of a cationic hydrogel, RL-100, with PtPFPP molecules embedded in the core; (**C**) small molecule derivatives of PtPFPP: tetra-PEGylated, PtPEG_4_ and tetra-glucosylated PtGlc_4_.

**Figure 3 sensors-24-00267-f003:**
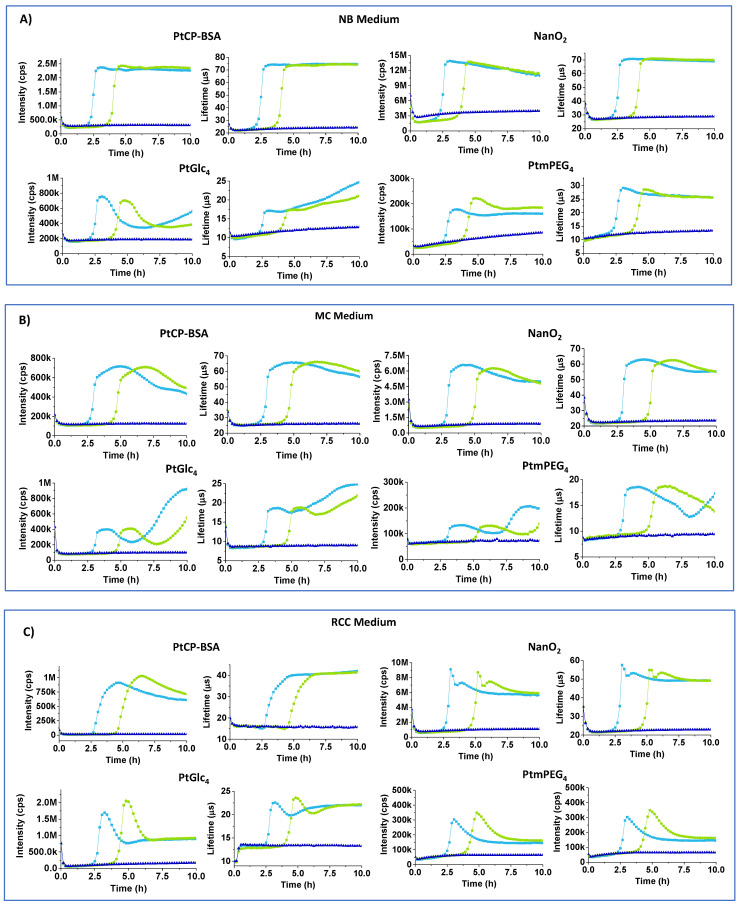
Intensity (cps) and LT (µs) profiles of MitoXpress (0.3 µM), NanO2 (0.3 µM), PtGlc_4_ (3 µM), PtPEG_4_ (3 µM), in NB (**A**), MC (**B**), RCC (**C**), MMG (**D**), and MLS (**E**) media with 10^4^ (green curves) and 10^6^ (light blue) *E. coli* CFU/mL, at 37 °C. Negative controls (probe in sterile media, no cells) are also shown (dark blue).

**Figure 4 sensors-24-00267-f004:**
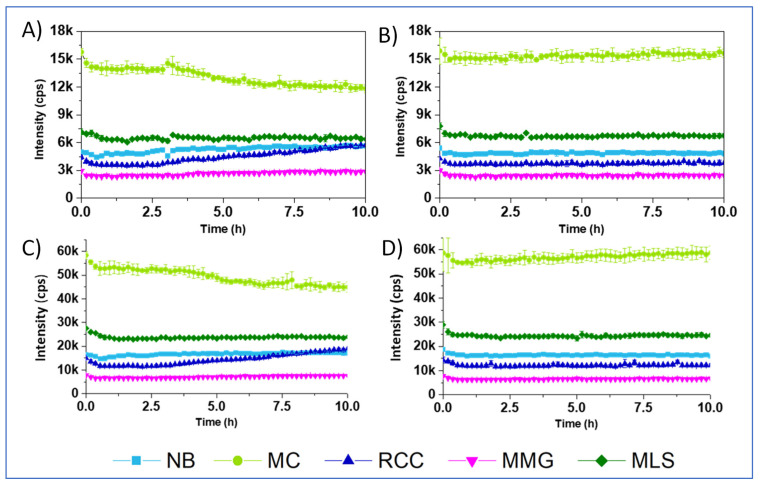
Profiles of blank Int signals (no probe) for the different media, recorded at delay times of 30 and 70 µs (**A**,**B**), or 25 and 50 µs (**C**,**D**), with 10^6^
*E. coli* CFU/mL (**A**,**C**) and without *E. coli* (**B**,**D**).

**Figure 5 sensors-24-00267-f005:**
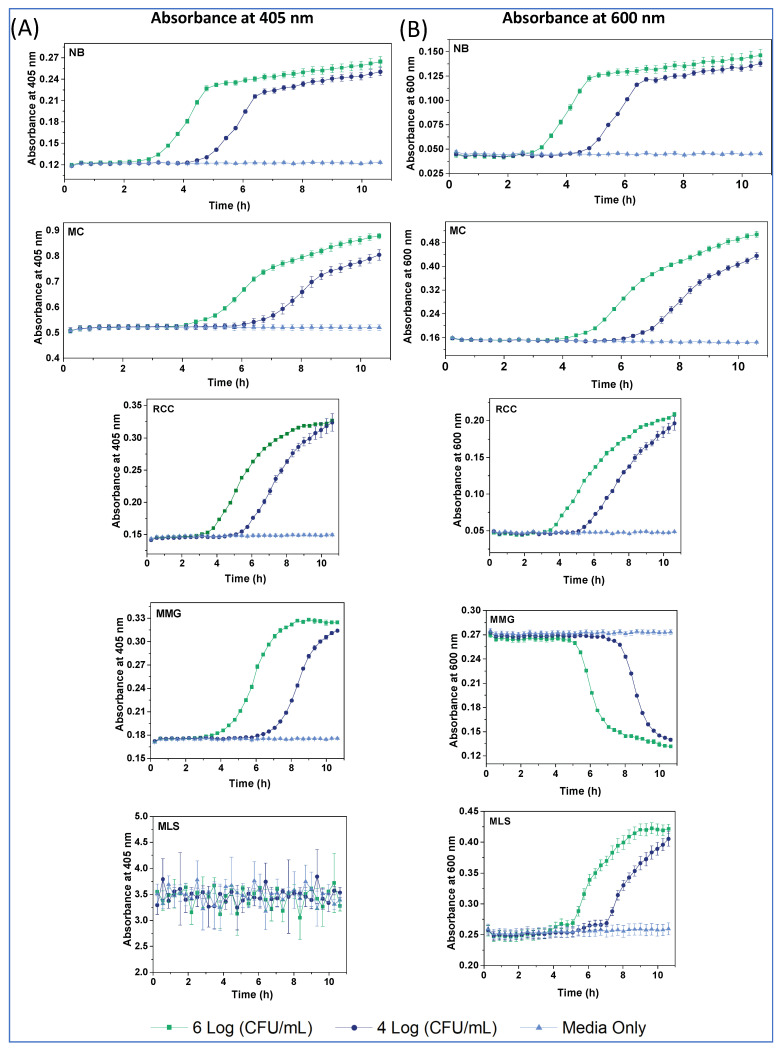
Densitometry profiles at 405 nm (**A**) and 600 nm (**B**) for the different media, at *E. coli* concentrations of 10^4^ (dark blue), 10^6^ (green) and zero (light blue) CFU/mL, and at 37 °C.

**Figure 6 sensors-24-00267-f006:**
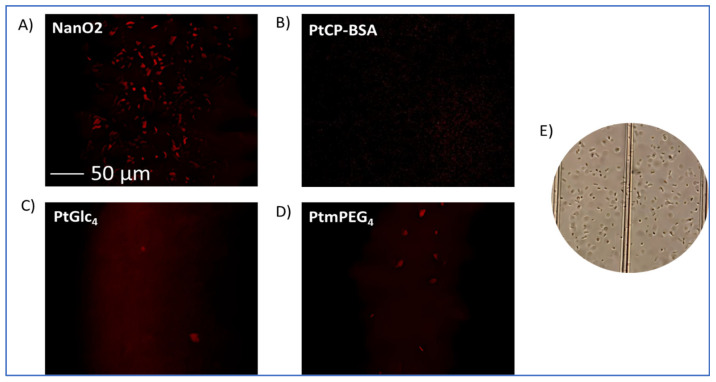
Phosphorescence intensity images of *E. coli* cells, pre-incubated for 3 h at 37 °C with NanO2 (**A**), MitoXpress/PtCP-BSA (**B**), PtGlc_4_ (**C**), and PtPEG_4_ (**D**) at working concentrations and measured on a confocal fluorescence microscope. (**E**) Transmission image of *E. coli* cells in NB broth in a haemocytometer.

**Table 1 sensors-24-00267-t001:** The O_2_ probes used in this study and their characteristic features.

Probe	Porphyrin Phosphor	Working Dilution	Probe Type	Special Features
**MitoXpress**	PtCP	0.3 µM	Macromolecular conjugate	Protein based, cell-impermeable
**NanO2**	PtPFPP	0.3 µM	Nanoparticle	Polymeric core–shell NPs, shielded, cell-permeable *
**PtGlc_4_**	Modified PtPFPP	3 µM	Small molecule	Cell-permeable *
**PtPEG_4_**	Modified PtPFPP	3 µM	Small molecule	Cell-impermeable *

* Assessed mostly mammalian cells and tissues.

**Table 2 sensors-24-00267-t002:** Basal and maximal phosphorescence signals of the probes in the different media.

Probe	Medium	I_base_ (cps)	I_max_ (cps)	LT_base_ (µs)	LT_max_ (µs)	Comments
**PtCP-BSA**	**NB** MC RCC MMG MLS	279k 110k **14k** 56.9k **22k**	2.4M 715k 908k 424k **120k**	22.1 25.2 **15.6** 19.6 **18.8**	74.6 65.3 **40.5** 46.7 **35.0**	Normal sigmoid, stable signals Altered sigmoid Almost normal sigmoid Bell-shaped Int, sigmoidal LT profiles Bell-shaped sigmoid
**NanO2**	**NB** MC RCC MMG MLS	1.7M 717k 738k 259k 530k	13.8M 6.5M 9.1M 2.1M 2.9M	27.1 22.4 21.8 **14.7** 26.4	70.6 62.8 57.7 **32.9** 62.5	Normal sigmoid, stable signals Almost sigmoidal profiles Almost sigmoidal profiles Sigmoidal profiles, shorter LT Altered sigmoid with downward trend
**PtGlc_4_**	**NB** MC RCC MMG MLS	164k 86k 75k 36k **29k**	756k 397k 1.7M 230k **43k**	** 10.3 ** ** 8.7 ** ** 13.3 ** ** 11.3 ** ** 9.2 **	**17.1** 18.5 22.6 17.3 **12.1**	Bell-shaped Int and sigmoidal LT with upward drift Altered profiles with two subsequent signal growth Signal drop followed by a plateau Altered sigmoid with upward trend Strongly quenched signals
**PtPEG_4_**	**NB** MC RCC MMG MLS	**36k** 69k 53k **17k** **41k**	178k 133k 303k 66k 109k	** 11.9 ** ** 8.8 ** ** 13.1 ** ** 10.9 ** ** 8.2 **	29.1 18.5 27.5 20.9 **11.8**	Almost sigmoidal profiles Low Int change and altered LT profile Bell shaped Int and almost sigmoidal LT Almost sigmoidal signals, lower amplitude Signal quenching

I_base_ = basal intensity; I_max_ = maximal intensity; cps = count per second; LT or τ = lifetime (µs). Maximal signals were obtained at [*E. coli*] = 10^6^ CFU/mL. Questionable signals are highlighted in red.

**Table 3 sensors-24-00267-t003:** Main parameters of the densitometry profiles (A_405_ and A_600_) of *E. coli* (at 10^6^ CFU/mL) in the different media at 37 °C, and comparison of their TT values with respirometry.

Media	Registration Wavelength	A_0_	A_max_	TT_D_, h	TT_R_, h
**NB**	405 nm 600 nm	0.12 0.04	0.26 0.15	2.18 2.83	2–2.2
**MC**	405 nm 600 nm	0.52 0.15	0.88 0.51	4.13 4.13	2.5–2.9
**RCC**	405 nm 600 nm	0.15 0.04	0.32 0.21	3.15 3.48	2.2–2.9
**MMG**	405 nm 600 nm	0.18 0.27	0.33 0.13	3.48 5.10	2.1–2.7
**MLS**	405 nm 600 nm	>3 0.25	n.m. 0.42	n.m. 4.13	2.2–2.9

A—absorbance (AU), TT_D_ and TT_R_—threshold times for densitometry and respirometry; n.m.—not measurable.

**Table 4 sensors-24-00267-t004:** Effects of the different components of the media on the O_2_ probes.

Ingredient	Concentration and Media Used	Changes in A_390_/A_650_, %/% at 1x	Quenching of Probe Int and LT signals %/% at 1x		
			NanO2	MitoXpress	PtGlc_4_
Lactose	MMG—20 g/L MLS—30 g/L	0/0	9.1/3.9	−13.7/0	−25.7/0
L-cystine	MMG—0.04 g/L	0/0	1.5/0	−1.0/4.5	0.9/0
Thiamine	MMG—0.002 g/L	0/0	6.5/3.9	0/0	0.6/5.6
Pantothenic acid	MMG—0.002 g/L	0/0	8.1/3.7	−1.8/4.3	2.3/0
Nicotinic acid	MMG—0.002 g/L	0/0	−1.4/4	0/0	5.3/0
Bile salts	MC—5.0 g/L	0/0	−31.1/0	−62.0/4.8	−825/55
Ferric ammonium citrate	MMG—0.02 g/L	<0.1	3.9/0	3.9/12	8.8/6.7
Lab-Lemco	NB—1.0 g/L	0.02/0	−16.3/0	−6.6/0	−17.9/0
Peptone	NB—5.0 g/L MC—20 g/L RCC—5.0 g/L	0.15/0	−1.6/0	−54.4/0	−281/20
Yeast extract	NB—2.0 g/L MLS—6 g/L	0.5/0	6.1/3.5	−116/4.8	−436/6.3
Sodium lauryl sulphate	MLS—1 g/L RCC—0.1 g/L	0/0	3.5/16	21.0/23.8	−268/50
X-gal	RCC—0.08 g/L	0/0	33.2/3.5	42.9/0	45.0/0
MUG	RCC—0.05 g/L	0/0	15.9/3.7	17.5/4.2	−16.1/6.7
IPTG	RCC—0.1 g/L	0/0	6.2/0	−8.4/8.3	1.7/6.7
Neutral red	MC—0.075 g/L	0.2/0.05	n.m. (IFE)	n.m. (IFE)	n.m. (IFE)
Phenol red	MLS—0.2 g/L	>3.0/0	n.m. (IFE)	n.m. (IFE)	n.m. (IFE)
Bromocresol purple	MMG—0.02 g/L	0.5/0	47.8/12.0	85.9/31.8	44.7/12.5

n.m.—not measurable; IFE—inner filter effect. Significant effects are shown in red (negative values correspond to signal enhancement).

**Table 5 sensors-24-00267-t005:** Summary of the general usability of the probes in the different growth media.

Probe	NB	MC	RCC	MMG	MLS
**MitoXpress**	+	+	-	+/-	-
**NanO_2_**	+	+	+/-	+	+
**PtmPEG_4_**	+/-	-	+/-	-	-
**PtGlc_4_**	+/-	+/-	+/-	-	-

## Data Availability

Data are available on request from the corresponding author.

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
