# Peer review of "Susceptibility of the Different Oxygen-Sensing Probes to Interferences in Respirometric Bacterial Assays with Complex Media"

_sensors, 2024, doi:10.3390/s24010267_

Round 1
Reviewer 1 Report
Comments and Suggestions for Authors
In this manuscript, the authors investigated the susceptibility of the different oxygen-sensing probes to interferences in respirometric bacterial assays with complex media. Typically, the macromolecular MitoXpress, nanoparticle NanO2 and small molecule PtGlc4 and PtPEG4 probes were assessed with E. coli cells in five growth media. The research value of this study is of great significance for sensing probes. Throughout the full text, I recommend its publication after minor revision.
Points for further revision:
1. About the involved four different water-soluble O2 sensing probes, the key factors for individual performance should be discussed comparatively in detail.
2. About the investigation of cell penetration and staining ability of the probes, why “only NanO2 probe provided efficient phosphorescent staining and visualization of the cells. The other probes showed no cell staining or very few stained cells…… ” ? The reason should be explained in detail.
3. Throughout the whole process of investigation in this manuscript, the practical application of these four probes should be indicated and further analyzed, especially the comparative results of this study.
Comments on the Quality of English LanguageModerate editing of English language required
Author Response
Query 1. About the involved four different water-soluble O2 sensing probes, the key factors for individual performance should be discussed comparatively in detail.
Answer 1. These factors are actually discussed in sufficient detail in sections 3.2 (pp. 4,5), 3.3 (pp. 5-7), 3.7 (pp. 16-17), and in Conclusions.
Q2. About the investigation of cell penetration and staining ability of the probes, why “only NanO2 probe provided efficient phosphorescent staining and visualization of the cells. The other probes showed no cell staining or very few stained cells…… ” ? The reason should be explained in detail.
A2. Cell penetrating ability of different O2 probes is determined by many factors, which relate to the cell, medium and compound in question. Such factors have been investigated in detail, but mainly with mammalian cells. Thus, positive charges and saccharide substituents (e.g. glucose, galactose) on the compound/probe tend to promote compound internalization [11,12,38], whereas PEGylation prevents it [14]. However, these are not universal rules, and for each system theoretical predictions usually require tedious experimental studies (trial and error) to optimize probe uptake and elucidate the mechanisms of internalization. Clearly, cell-penetrating behavior of an O2 probe in mammalian cells cannot be simply extrapolated on bacterial cells, which have very different membrane structure, transport mechanisms, growth rates, etc. Therefore, direct phosphorescent staining and imaging experiments were conducted with E. coli cells, to determine cell-staining behavior of each probe. Unlike with mammalian cells [11], PtGlc4 did not show good staining of E.coli, so it behaved as cell impermeable probe. On the other hand, the intracellular behavior of NanO2 with E. coli, and extracellular behavior of the MitoXpress and PtPEG4 were similar to that of the mammalian cells (see Table 1, [12,13]).
This is now explained in the revised MS (p.15)
Q3. Throughout the whole process of investigation in this manuscript, the practical application of these four probes should be indicated and further analyzed, especially the comparative results of this study.
A3. Again, this can be found in section 3.7 and Conclusions (pp.15-17).
Reviewer 2 Report
Comments and Suggestions for Authors
The research has definite practical impact, it is well designed, implemented and described. The manuscript is good and may be published. There are no significant points of criticism. Minor points are listed below and in "comments on English".
Section 3.5 and figure 5. Please double check the text for correct use of “absorption” and “absorbance” (absorption+scattering)
For discussion: E. coli is a typical facultative anaerobe. When you grow them in liquid medium without shaking they start as aerobic and consume dioxygen, producing the sought-for rise in the TR-F signal. As dioxygen drops below certain level, the cells switch to glycolysis (downregulating respiration chains, decreasing oxygen consumption per cell), and begin to acidify the medium. It causes changes in absorption of the indicators, as well as poorly defined absorbing components in peptone etc, it is proved by absorption kinetics. The decrease in pH noticeably lags behind the drop of O2 (as H+ need time to accumulate). This fact may relate to the transient peaks in the response curves in Fig. 3. Aerobic E. coli growth also actively consumes iron (which, as you have noted, can quench) and copper from the medium, while anaerobic growth needs them in small amounts. Also, in anaerobic growth in rich media, cells can start to release LMW thiols (H2S, CH3SH), which may interfere with platinum sensors.
After examining your excellent research I see that PTFE-covered Clark electrode (despite its hugeness) is still a competitive technique in some challenging media.
Comments on the Quality of English LanguageLines 148 and 158. Check “specie” : probably a misprint (“species” is singular and plural)
Line 169. “mammalians”, probably should be “mammalian”
Author Response
Q1. Section 3.5 and figure 5. Please double check the text for correct use of “absorption” and “absorbance” (absorption+scattering)
A1. Implemented.
Q2. For discussion: E. coli is a typical facultative anaerobe. When you grow them in liquid medium without shaking they start as aerobic and consume dioxygen, producing the sought-for rise in the TR-F signal. As dioxygen drops below certain level, the cells switch to glycolysis (downregulating respiration chains, decreasing oxygen consumption per cell), and begin to acidify the medium. It causes changes in absorption of the indicators, as well as poorly defined absorbing components in peptone etc, it is proved by absorption kinetics. The decrease in pH noticeably lags behind the drop of O2 (as H+ need time to accumulate). This fact may relate to the transient peaks in the response curves in Fig. 3. Aerobic E. coli growth also actively consumes iron (which, as you have noted, can quench) and copper from the medium, while anaerobic growth needs them in small amounts. Also, in anaerobic growth in rich media, cells can start to release LMW thiols (H2S, CH3SH), which may interfere with platinum sensors.
A2. These are valid points and useful hints. We will try to investigate them in our future studies.
Q3. After examining your excellent research I see that PTFE-covered Clark electrode (despite its hugeness) is still a competitive technique in some challenging media.
A3. We agree with this statement, while keeping in mind that solid-state phosphorescent O2 sensors also have ‘shielding’ effect of the polymer matrix [9,12] and are resilient to challenging conditions.
Q4. Lines 148 and 158. Check “specie” : probably a misprint (“species” is singular and plural). Line 169. “mammalians”, probably should be “mammalian
A4. Now corrected
Reviewer 3 Report
Comments and Suggestions for Authors
This was a good paper that focused on the advantages and disadvantages of the four different oxygen probes, as well as the differences in usage (ph, media composition, etc.).
[Major]
Although the authors compared each oxygen probe, it should be compared to other oxygen sensors to assess if the oxygen concentrations obtained from the probes are correct.
Author Response
Q1. Although the authors compared each oxygen probe, it should be compared to other oxygen sensors to assess if the oxygen concentrations obtained from the probes are correct.
A1. Bacterial cell respirometry normally deals with primary sensor signal (Int or lifetime) [3]. LT mode provides reliable determination of TT values (hours) for tested samples. O2 concentration readout is not advantageous, as it is more prone to measurement errors and artefacts than LT readout. Especially in the challenging conditions like here.
Our O2 probes were selected based on their compatibility with the common detection platform: standard microplates and TRF reader. This panel of probes is diverse and representative. Other known probes are either not compatible with the TRF reader platform, or not available for this study. In the previous study (Ref. [9]), the MitoXpress platform was compared with the platform that uses vials with solid-state sensor coatings and a handheld sensor reader.
This was explained in Introduction (p.2) and Results (sections 3.4 and 3.7).
Reviewer 4 Report
Comments and Suggestions for Authors
Authors evaluated comparatively four different O2-sensing probes based on Pt-porphyrin phosphors in respirometric bacterial assays performed on a time-resolved plate reader. They claimed that their goal was to identify the best performing probe for such assays and elaborate the particular causes of interferences on probes’ phosphorescent signals and the resulting respiration profiles.
While the experimental work and analytical characterization of the results are overwhelming, the following points are concerning:
1. 50 % of self-citation is extremely high and I find it unacceptable
2. I do not find a significant novel work provided in the current study taking in consideration the work presented in references [4] and [15].
3. The quality of the English language can be improved.
Therefore, I decided to reject the manuscript.
[15] https://doi.org/10.1016/j.snb.2023.133582
[4] https://doi.org/10.3390/s23094519
Comments on the Quality of English LanguageI suggest a native English speaker to read the entire manuscript and edit accordingly.
Author Response
Q1. 50 % of self-citation is extremely high and I find it unacceptable
A1. We have addressed this issue and reduced the number of self-citations by 10. This gives a better balance, also considering the relatively small number of publications on this subject. Further reduction would not be practical and justifiable.
Q2. I do not find a significant novel work provided in the current study taking in consideration the work presented in references [4] and [15].
A2. Ref [4] (now [3]) is a general review, it does not contain any detail related to this study. In Ref [15] (now [9]), different respirometric platforms was compared, rather than different probes. While this study uses the same cell and media models as [15], its scope is much broader and deeper and contains new mechanistic insights and multiple novelty elements.
All this is described in Introduction, Conclusions and Abstract.
Reviewer 5 Report
Comments and Suggestions for Authors
General comments
The study is aimed at examining the applicability of the phosphorescence-based probes of various molecular sizes and solubilities in respirometric assays of bacteria growing in different standardized media. An impressive number of experiments was carried out to evaluate the contributions of the media effects like reabsorption, autofluorescence and quenching into response signals of the probes. The final conclusion was made that the best probe is one with the phosphor protected from the contacts with a medium by polymeric matrix and possessing the highest brightness. Surprisingly, for “the best” probe NanO2 the possibility to produce adequate response signals in a medium with an optical density at the excitation wavelength >3 was shown.
From my point of view there are no questions to the subject of this study, to the methods and results. But the form of presentation requires a lot of corrections, especially the figures and the tables, which are rather sloppily compiled.
Specific comments
1. Many inaccuracies and minor errors should be corrected:
1.1) E.coli is not in italic somewhere (Lines 62, 288 and may be others)
1.2) Micromolar concentration is designated uM (Line 137), in many cases the space is missed between the value and the unit (Lines 99, 100, 105, 136, 308 and others)
1.3) Line 465: “Amax 615 nm” should be “Amax at 615 nm” or “lmax is 615 nm” (?)
2. The quality of the figures and tables should be improved:
2.1) in Fig. 2, B and C, the inscriptions are not readable
2.2) In Fig. 3 and 5 the titles of the axes are in too small a font. They are also hardly readable.
2.3) In Fig. 5 the panels A and B are not designated.
The concentration of bacterial cells in the legend should be shown in explicit form (106 and 104). As well as in line 238.
I would recommend organizing each panel vertically (all medium one under the other). Thus, the left side of the figure will be panel A, and the right side will be panel B.
2.4) Line 236, Table 2:
- I would recommend presentation of the intensity values in exponential form like 103 and 106, not like k and M. As well as in discussion elsewhere in the text (lines 242, 245, 246 and others).
- Line 237: TT should be removed; it is not mentioned in the table.
- Ip in the last column is not explained anywhere.
2.5) Line 413, Table 3:
- The concentration of the bacteria (106) should be mentioned in the Table caption, not in the table footter, because it is relevant to all the data in the table.
- Designations l405 and l600 are not correct in the places where they are. A405 and A600 should be or simply 405 and 600, if you write “Registration wavelength, nm” in the column header.
-“n.m.“ should be explained in the footnotes.
2.6) Line 453: Table 4:
- Why some symbols are in bold, some – in red, some – in bold red? Impossible to guess and no information about it.
- The names of the last three ingredients are written from lower case unlike the others. What does it mean?
- Table footter: “* - signal enhancement” is indicated, but no “*” could not be found in the table.
- The headers: Abs390/A650 probably should be A390/A650 (?). What the “0” mean in this column? Is it “0/0”? Please unify the data in this column.
3. Other comments:
3.1) The pulse duration of Xe-flash lamp should be indicated in the methods section (part 2.3). It helps understanding why some values in Table 2 are “questionable”.
3.2) In lines 104-114 one should add information about how the "TR-F intensity" is obtained in such a time-resolved mode with two delay times and gate time.
3.3) I recommend replacing “TR-F Int” in discussion (lines 248, 261, …) and in figures (Fig. 3 and 4) by “Intensity” or “Int” (which is used in lines 293, 297…), because in fact it is intensity of phosphorescence, not fluorescence.
3.4) Lines 434 and 556: “solution FRET” sounds strange. It is my understanding that there is a special type of the experimental technique called “in solution” FRET, but the mechanism of luminescence quenching is FRET. I would recommend removing “solution”.
3.5) The manuscript appears to have a high level of self-citation. Among the 19 references in the Introduction, 15 are self-citations. Even the composition of the standard medium NB (Line 291) have the reference [15], which is of the manuscript’s authors. I would recommend citing more papers by other authors.
3.6) Lines 155, 368: Some supporting information is mentioned, but I could not download any ESI file for this manuscript.
Author Response
- Many inaccuracies and minor errors should be corrected:
1.1) E.coli is not in italic somewhere (Lines 62, 288 and may be others)
A. Corrected
1.2) Micromolar concentration is designated uM (Line 137), in many cases the space is missed between the value and the unit (Lines 99, 100, 105, 136, 308 and others)
A. Corrected
1.3) Line 465: “Amax 615 nm” should be “Amax at 615 nm” or “lmax is 615 nm” (?)
A. Corrected
- The quality of the figures and tables should be improved:
2.1) in Fig. 2, B and C, the inscriptions are not readable
A. Improved
2.2) In Fig. 3 and 5 the titles of the axes are in too small a font. They are also hardly readable.
A. Corrected
2.3) In Fig. 5 the panels A and B are not designated.
A. Corrected
The concentration of bacterial cells in the legend should be shown in explicit form (106 and 104). As well as in line 238.
A. Now corrected
I would recommend organizing each panel vertically (all medium one under the other). Thus, the left side of the figure will be panel A, and the right side will be panel B.
A. Implemented
2.4) Line 236, Table 2:
- I would recommend presentation of the intensity values in exponential form like 103 and 106, not like k and M. As well as in discussion elsewhere in the text (lines 242, 245, 246 and others).
A. Proposed format is more bulky and difficult to read (and fit). Authors think that such changes are unnecessary.
- Line 237: TT should be removed; it is not mentioned in the table.
A. Done
- Ip in the last column is not explained anywhere.
A. Corrected - Int
2.5) Line 413, Table 3:
- The concentration of the bacteria (106) should be mentioned in the Table caption, not in the table footter, because it is relevant to all the data in the table.
A. Done
- Designations l405 and l600 are not correct in the places where they are. A405 and A600 should be or simply 405 and 600, if you write “Registration wavelength, nm” in the column header.
A. Corrected
-“n.m.“ should be explained in the footnotes.
A. Done
2.6) Line 453: Table 4:
- Why some symbols are in bold, some – in red, some – in bold red? Impossible to guess and no information about it.
A. Corrected
- The names of the last three ingredients are written from lower case unlike the others. What does it mean?
- Table footter: “* - signal enhancement” is indicated, but no “*” could not be found in the table.
A. Explained and corrected
- The headers: Abs390/A650 probably should be A390/A650 (?). What the “0” mean in this column? Is it “0/0”? Please unify the data in this column.
A. Corrected and clarified
- Other comments:
3.1) The pulse duration of Xe-flash lamp should be indicated in the methods section (part 2.3). It helps understanding why some values in Table 2 are “questionable”.
A. Flash duration is an inherent characteristic of the light source used in such instruments (Xe-flash lamp), users cannot control or change it. So, authors do not agree with this suggestion.
3.2) In lines 104-114 one should add information about how the "TR-F intensity" is obtained in such a time-resolved mode with two delay times and gate time.
A. This is standard algorithm/ measurement mode is described in many papers, references to which are given.
3.3) I recommend replacing “TR-F Int” in discussion (lines 248, 261, …) and in figures (Fig. 3 and 4) by “Intensity” or “Int” (which is used in lines 293, 297…), because in fact it is intensity of phosphorescence, not fluorescence.
A. Agreed and corrected
3.4) Lines 434 and 556: “solution FRET” sounds strange. It is my understanding that there is a special type of the experimental technique called “in solution” FRET, but the mechanism of luminescence quenching is FRET. I would recommend removing “solution”.
A. Agreed and removed
3.5) The manuscript appears to have a high level of self-citation. Among the 19 references in the Introduction, 15 are self-citations. Even the composition of the standard medium NB (Line 291) have the reference [15], which is of the manuscript’s authors. I would recommend citing more papers by other authors.
A. We have removed 10 self references and adjusted citations in the text. The balance has improved now .
3.6) Lines 155, 368: Some supporting information is mentioned, but I could not download any ESI file for this manuscript.
A. Web link to ESI has been verified, it should work now
Round 2
Reviewer 4 Report
Comments and Suggestions for Authors
The quality of the manuscript has been improved compared with the original submission. I now find the manuscript fit to be published in 'Sensor'.
Author Response
We thank Reviewer 4 for the positive assessment of our revised manuscript.
Reviewer 5 Report
Comments and Suggestions for Authors
I think the quality of presentation of results in the manuscript has improved significantly. Most of the comments were taken into account during revision.
However, I also recommend correcting some minor issues:
1) Line 165: The units for brightness must be specified.
2) Figure 3 (lines 318 and 320): D and E are absent in the last panels of the figure.
3) Figure 4 (Lines 379-380): The panels in the figure are a), b) c) and d), but in the figure caption they are A, B, C and D.
4) Figure 5 (line 401): the cells concentration in the legend should be written in different way, like in the rest of the manuscript (104 and 106).
5) Table 4 (line 454): Again, it is unclear what "0" or "<0.1" in the third column means, because according to the header there should be two numbers separated by a "/" (like 0.02/0 or 15/0). The correction is needed.
Author Response
The requested corrections have now been implemented:
1) Line 165: The units for brightness must be specified.
Done, see l.165
2) Figure 3 (lines 318 and 320): D and E are absent in the last panels of the figure.
These labels have now been provided.
3) Figure 4 (Lines 379-380): The panels in the figure are a), b) c) and d), but in the figure caption they are A, B, C and D.
These corrections have been implemented
4) Figure 5 (line 401): the cells concentration in the legend should be written in different way, like in the rest of the manuscript (104 and 106).
Now corrected
5) Table 4 (line 454): Again, it is unclear what "0" or "<0.1" in the third column means, because according to the header there should be two numbers separated by a "/" (like 0.02/0 or 15/0). The correction is needed.
The missing numbers have now been written (highlighted in yellow